# Efficient Embedding-Generation Serving with Heterogeneous Batching

## Abstract

Modern information retrieval increasingly relies on both embedding and generative models to achieve high accuracy. To make such applications more responsive, the underlying serving systems must be optimized for mixed workloads. Yet, current systems suffer from low throughput and poor GPU utilization, primarily because they cannot batch embedding and generation requests together. We address this bottleneck with heterogeneous batching, which schedules embedding and generation requests within the same batch. Realizing this idea requires two changes to the system internals: a *unified kernel abstraction* and *fine-grained intra-batch scheduling*. The unified abstraction enables concurrent handling of embedding and generation, while the intra-batch scheduler dynamically adapts batch composition to balance end-to-end throughput across both tasks. Our evaluation with four A100 GPUs shows that heterogeneous batching achieves $1.28\times$-$4.52\times$ higher throughput and 35.8-52.0% lower latency than default vLLM.

## 1 Introduction

Modern information retrieval (Asai et al., 2023; Gao et al., 2023; Jiang et al., 2023b; Shao et al., 2023) employs both embedding and generative models for improved accuracy. A query is first *embedded* for dense retrieval, and then a subsequent query is *generated* capturing missing information. This iteration continues until sufficient information is retrieved. Accordingly, search becomes more responsive when the model serving is efficient for mixed embedding and generation workloads.

Yet, existing systems provide low throughput and poor GPU utilization for mixed workloads. Serving systems (e.g., Kwon et al. (2023)) run each model separately. With multiple GPUs, some deployments can handle embeddings and others generation, but this causes two problems: (i) predicting the right embedding-generation ratio is hard, and (ii) changing the split requires costly model reloading. Even with perfect knowledge of future workloads, GPU-level split delivers suboptimal performance as embedding servers are compute-bound and generation servers are memory-bound. We hypothesize that homogeneous batching is a fundamental bottleneck in leveraging GPU resources efficiently.

In this work, we present a systems approach for efficiently serving mixed embedding-generative workloads. Our key idea is to unify the two distinct compute patterns—embedding and generation—into a single abstraction, enabling heterogeneous batching within a single inference step. Embedding and generation kernels both inherit this abstraction and access their respective parameters. This design allows fine-grained load balancing and improved GPU utilization, yielding substantially higher throughput. It requires no additional training and applies broadly to existing models.

Designing our system involves two high-level considerations: (i) unified abstraction and (ii) heterogeneous batching. We pursue high throughput and optimal intra-batch scheduling as follows:

- *Unified abstraction*: Our abstraction $\kappa$ takes the current state and tokens $(\phi_{t-1}, \tau_t)$ to produce a new state $\phi_t$, where $\phi_t$ represents data retained in GPU global memory. This abstraction implies that tokens are processed in a streaming fashion. Nearly all existing embedding kernels violate this pattern. Our new embedding kernel adheres to it with *incremental pooling*.[1]

---

[1] We have noticed that the latest version of vLLM, developed concurrently, introduced a similar incremental strategy to handle long context.

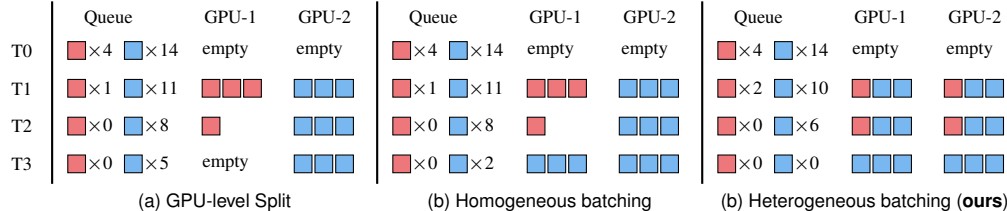

Figure 1: Example embedding-generation workload processing. Workload is given at T0. Both (a) GPU-level split and (b) homogeneous batching on a combined model cannot complete all requests by T3. Our proposed heterogeneous batching completes all 18 units of workloads (4 + 14).

- **Optimal batching**: Our heterogeneous batching aims to balance the end-to-end throughput between embedding and generation under dynamic workloads. Our system achieves higher throughput when compared to every possible GPU-level split.

Fig. 1 demonstrates a mixed workload for which heterogeneous batching can be more efficient than other options. A mixed workload is submitted at time 0 (T0), where the number of red and blue boxes in the Queue indicates the amount of remaining workload. GPU-level split, with some GPUs dedicated to embedding and others to generation, cannot utilize GPU resources if the workload doesn't perfectly match a pre-determined split. Likewise, homogeneous batching cannot fully utilize available GPU memory (at T2). Our proposed heterogeneous batching completes the workload by fully utilizing GPUs' compute and memory resources. We analyze this formally in Section 3.

Our technique is practical and delivers high performance. We implemented our prototype system (called ORTHRUS) on top of vLLM version 0.92, without modifying memory management or model specification. Evaluation on four NVIDIA A100 GPUs with state-of-the-art open-weight models shows ORTHRUS delivers $1.28\times$-$4.52\times$ higher normalized throughput[2] than GPU-level splits, without any approximation on the inference results, for mixed embedding and generation workloads.

This work is orthogonal to memory-efficient model composition. Specifically, embedding and generative models can share a large fraction of parameters through LoRA adapters (Vasani et al., 2025; Gauthier, 2024) or by training a combined model (Muennighoff, 2022; Zhang et al., 2025; Springer et al., 2024; Li et al., 2024). While this work benefits from these advances, we advance compute patterns inside GPU kernels. This work is also orthogonal to efficiently scheduling multiple LoRA adapters (Sheng et al., 2024a; Chen et al., 2024); we focus on designing an effective kernel for heterogeneous batching rather than combining existing ones.

In summary, this work makes the following technical contributions:

1. To our knowledge, we are the first to propose, design, and implement heterogeneous batching of embedding and generation requests within each inference step.
2. We formalize our approach with a unified runner abstraction and throughput-aware intra-batch scheduling. We have open-sourced our system including evaluation scripts and data.[3]
3. We evaluate our system (ORTHRUS) from multiple angles to demonstrate higher throughput, reduced latency, instant load balancing, low overhead, and generalization to existing models.

The rest of this paper is organized as follows. Section 2 discusses our technical novelty in relation to existing literature, Section 3 presents our system, and Section 4 evaluates our system. Finally, Section 5 summarizes this work.

## 2 RELATED WORK

### 2.1 EMBEDDING AND GENERATION FOR MODERN INFORMATION RETRIEVAL

Knowledge-intensive tasks increasingly combine embedding and generative models in iterative pipelines that refine retrieval and improve answer quality. Embedding models capture dense semantic similarity for document retrieval, while generative models synthesize or reformulate queries

---

[2]*Normalized throughput* rescales throughput so embedding and generation are comparable (see Section 4.1).

[3]Our system is open-sourced at `https://anonymous.4open.science/r/Orthrus-483B`.

and answers (Gao et al., 2023; Wang et al., 2023; Shen et al., 2024). Retrieval-Augmented Generation (RAG) (Lewis et al., 2020) grounds generation in retrieved evidence, and recent work extends this paradigm with iterative retrieval and reasoning (Asai et al., 2023; Jiang et al., 2023b; Shao et al., 2023; Trivedi et al., 2023; Lyu et al., 2024). These methods highlight the effectiveness of embedding–generation loops, but also demand efficient inference for mixed workloads. Our system addresses this challenge by improving inference efficiency and responsiveness for such pipelines.

## 2.2 EFFICIENT PARAMETER SHARING BETWEEN EMBEDDING AND GENERATIVE MODELS

State-of-the-art embedding models are increasingly derived from generative models via parameter-efficient fine-tuning, most commonly with LoRA. Models such as GTE-Qwen2-7B-instruct (Li et al., 2023), E5-Mistral-7B (Wang et al., 2024), SFR-Embedding-Mistral (Meng et al., 2024), RepLLaMA-7B-Passage (Ma et al., 2024), and LLM2Vec (BehnamGhader et al., 2024) fine-tune base models like Qwen2, Mistral, or LLaMA, reusing nearly all parameters while adding lightweight adapters. This trend shows that embedding and generative models share substantial weights, with task-specific behavior introduced through adapters.

Beyond fine-tuning, several works train models to support both capabilities directly. Adapting generative transformers (Muennighoff et al., 2024), using transformers without modification (Muennighoff, 2022), or introducing summary tokens as embeddings (Zhang et al., 2025). Other methods derive embeddings from repeated inputs (Springer et al., 2024) or jointly optimize retrieval and generation with aligned representations (Li et al., 2024; Tang et al., 2025).

Together, these approaches demonstrate the feasibility of parameter sharing across embedding and generation tasks, but they focus on model training and design. The equally important challenge of efficiently serving such unified models at inference remains largely unexplored. Our work addresses this gap by enabling efficient serving of embedding and generation workloads under mixed demands.

## 2.3 LACK OF HETEROGENEOUS BATCHING IN SERVING SYSTEMS

Existing serving systems are optimized for generative workloads, structuring scheduling and batching around the autoregressive decode loop. A common baseline is to process requests in First-Come-First-Serve (FCFS) order or enforce client rate limits, which simplifies batching but ignores workload diversity. More advanced systems improve on this baseline in different ways: Orca (Yu et al., 2022) introduced *iteration-level scheduling*, running one decode step per iteration so requests can enter or leave dynamically. vLLM (Kwon et al., 2023) added *PagedAttention* for efficient KV-cache management. Sarathi-Serve (Agrawal et al., 2024) proposed *chunked prefill* to reduce stalls from large prompts. Other directions include *kernel fusion* (Wang et al., 2021; Fang et al., 2021), *memory offloading* (Sheng et al., 2023), *prefill/decode disaggregation* (Patel et al., 2024; Zhong et al., 2024; Hu et al., 2024), *preemptive scheduling* (Wu et al., 2024), and *fair scheduling* (Sheng et al., 2024b). Despite these advances, all assume decode-oriented workloads. Embedding requests, which involve only prefills, either wait behind long decodes or fragment batches when mixed naively.

Another line of work addresses multi-tenant serving by co-locating LoRA adapters. SLoRA (Sheng et al., 2024a) unified paging for adapter weights, and Punica (Chen et al., 2024) develops segmented gather kernels for batching LoRA variants. These systems improve throughput when serving many LoRA adapters, but still assume generation-only workloads. Crucially, embedding and generation require different *processing steps*. Generation performs sampling, whereas embedding pool hidden states into a fixed-length vector. Prior systems treat every request as a decode loop, forcing embeddings to be deployed as a separate service. Our system instead enables *heterogeneous batching*, efficiently co-serving embedding and generation workloads on the same infrastructure.

## 3 UNIFIED MODEL RUNNER FOR HETEROGENEOUS BATCHING

To utilize GPU resources more efficiently, we propose heterogeneous batching of embedding and generation requests. First, we describe our design for heterogeneous batching. Second, we introduce a mathematical model to show that heterogeneous batching can achieve higher throughput. Finally, we continue our formal analysis to study how we should compose each batch for low latency.

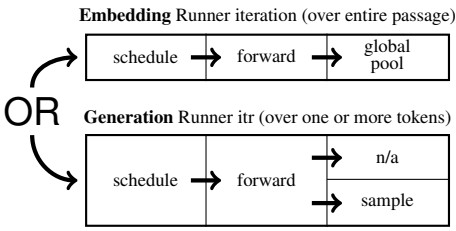

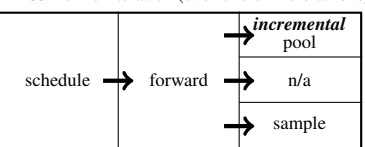

(a) Exclusive iterations for embedding/generation      (b) Unified iteration for heterogeneous batching

Figure 2: Runner structure comparison. The conventional approach presented on the left is designed for completely separate embedding and generation serving. Combining them is challenging due to the discrepancy between global aggregation vs token-level sampling. Our unified runner performs incremental pooling, allowing efficient concurrent processing of embedding and generation requests.

### 3.1 UNIFIED RUNNER FOR HETEROGENEOUS BATCHING

At a very high level, our runner is designed for iteration-level batching (Yu et al., 2022), aiming for seamless integration with the vLLM (Kwon et al., 2023)'s open-sourced model management and asynchronous server. Yet, our approach differs from existing approaches in that both embedding and generation requests are processed using the same runner, which we achieve by consolidating distinct embedding and generation runners into one in a structured way.

Our runner operates in three blocking stages: (1) `schedule`, (2) `forward`, and (3) `emit` (see Fig. 2 for illustration). `Schedule` determines batch composition, including generation prefills, decodes, or embedding chunks. `Forward` applies attention and feed-forward layers, identical across request types except for input length. `Emit` diverges: generation samples the next token, while embedding performs *incremental pooling* to maintain statistics for the final representation $\mathcal{P}(H)$. Because prefills are *chunked*, hidden states $H = \{H_t\}_{t=1}^{n}$ arrive incrementally; the aggregator avoids materializing all $H_t$ by keeping only minimal state. A running sum $S$ and count $n$ for mean pooling, the most recent $H_t$ for last-token pooling, or the full sequence if required. This ensures consistency under chunked execution while enabling concurrent embedding and generation.

**Implementation** The vLLM architecture consists of four components: an asynchronous server, a scheduler, model runners, and a memory manager based on PagedAttention. Our modifications focus on the scheduler and model runner. We introduce a *unified model runner* that combines the generative and pooling runners, executes the forward pass, and invokes sampler and pooler kernels in parallel. We also add a scheduling policy that constructs heterogeneous batches while remaining compatible with vLLM's execution model. On the kernel side, we implement Triton-based incremental pooling operators that run concurrently with sampling, and an embedding prefill kernel built on FlashAttention. Recent versions of vLLM support chunked prefill for embeddings with mean pooling to handle long prompts. Our design generalizes this to arbitrary pooling operators and integrates them into heterogeneous batching, allowing embeddings and generations to be processed concurrently. For embeddings, LoRA weights are pinned in GPU memory, with request indices passed to selectively apply low-rank updates.

### 3.2 THROUGHPUT ANALYSIS: HOMOGENEOUS VS HETEROGENEOUS BATCHING

We formally analyze the throughput difference between homogeneous and heterogeneous batching. This analysis indicates heterogeneous batching can achieve higher efficiency because embedding requests can be *squeezed* into the same batch containing generation requests. Unlike prior analytical models of LLM serving that focus on prefill/decode scheduling under FCFS assumptions (Yu et al., 2022; Agrawal et al., 2024), our formulation explicitly incorporates both embedding and generation workloads into the same iteration-based batch model.

A workload consists of $w_e$ embedding and $w_g$ generation requests. An embedding request takes $s_e$ steps to complete, and a generation request takes $s_g$ steps. We have $M$ memory; scheduling an embedding request consumes $m_e$ memory, and a generation request consumes $m_g$ memory. Let

$s_g = F \cdot s_e$, $m_g = F \cdot m_e$ and $M = L \cdot m_e$, where $F \geq 1$ captures the general phenomenon that generation requires more steps (for decoding) and memory (for KV cache).

Let $L = kF + r$ with $0 < r < F$ for some $k = \lfloor L/F \rfloor$. The number of steps $T_{homo}$ required by homogeneous batching for completing the workload is:

$$T_{homo} = \left\lceil \frac{w_e}{L} \right\rceil + \left\lceil \frac{Fw_g}{\lfloor L/F \rfloor} \right\rceil = \left\lceil \frac{w_e}{L} \right\rceil + \left\lceil \frac{F^2 w_g}{L - r} \right\rceil$$

Heterogeneous batching can leverage memory more effectively by filling in $r$ *slots* with embedding requests, as follows:

$$T_{hetro} = \left\lceil \frac{w_e + F^2 w_g}{L} \right\rceil$$

Heterogeneous batching can thus reduce the total number of steps as follows:

$$T_{homo} - T_{hetro} \approx \underbrace{\left( \left\lceil \frac{w_e}{L} \right\rceil + \left\lceil \frac{F^2 w_g}{L} \right\rceil - \left\lceil \frac{w_e + F^2 w_g}{L} \right\rceil \right)}_{\in [0,1]} + \underbrace{\left( \left\lceil \frac{F^2 w_g}{L - r} \right\rceil - \left\lceil \frac{F^2 w_g}{L} \right\rceil \right)}_{\text{extra penalty from } r \geq 0}$$

Heterogeneous batching is almost always more efficient than homogeneous batching, and the gap widens when generative requests produce long answers that require more memory, resulting in a larger $r$. This analysis is based on an assumption that is actually favorable to homogeneous batching: it presumes that adjusting the GPU split ratio incurs zero overhead and that context switching between embedding and generation is instantaneous. In practice, neither of these assumptions holds, which further increases the performance gap between homogeneous and heterogeneous batching.

## 3.3 INTRA-BATCH SCHEDULING (IBS) FOR LOW-LATENCY SERVING

With heterogeneous batching, any scheduling policy can deliver near-identical throughput; however, their latencies may differ. In this section, we analyze the optimal batch composition for minimizing the request-level latency. We aim to ensure "first come, first served" within each embedding/generation queue; however, some later generations may be scheduled before earlier embeddings (or vice versa) to reduce latency.

We have $w_e$ embedding and $w_g$ generation requests at time 0. Let $(x, y)$ be the number of embedding and generation requests scheduled within a batch. $x + Fy \leq L$. Set $t_e = 1$ without loss of generality.

- **Embeddings.** $x$ requests complete in each step (since $t_e = 1$). $j$-th embedding (in the queue) completes at $j/x$. The average embedding latency is then:

$$\bar{T}_e(y) \approx \frac{w_e}{2x} = \frac{w_e}{2(L - Fy)}$$

- **Generation.** We finish $y$ requests every $F$ steps (since $t_g = F \cdot t_e = F$). The $k$-th *group* of size $y$ thus completes at $kF$. The average generation latency is:

$$\bar{T}_g(y) \approx \frac{Fw_g}{2y}$$

The average latency across embedding and generation requests $\bar{T}(y)$ is a weighted mean of $\bar{T}_e(y)$ and $\bar{T}_g(y)$, expressed as:

$$\bar{T}(y) = \frac{w_e \bar{T}_e(y) + w_g \bar{T}_g(y)}{w_e + w_g} \quad \propto \quad f(y) = \frac{w_e^2}{L - Fy} + \frac{Fw_g^2}{y} \quad \rightarrow \quad y^* = \frac{Lw_g}{w_e + Fw_g}$$

where $y^*$ is obtained by setting $f'(y) = 0$. Finally, we get $x^* : y^* = w_e : w_g$, indicating the batch composition should be dynamically adjusted to match the respective queue size.

## 4 EVALUATION

We evaluate ORTHRUS, an embedding-generation serving system with heterogeneous batching. The setup is described in Section 4.1. Our results show that:

- **Throughput:** ORTHRUS improves average throughput under mixed embedding and generation workloads by $1.28\times$-$4.52\times$, compared to GPU-level model splits (Section 4.2).
- **Low Latency with Intra-batch Scheduling (IBS):** IBS improves request-level latency. ORTHRUS improves p99 generation latency by 9% compared to GPU level split baselines, and p99 embedding latency by 16% compared to naive scheduling baselines (Section 4.3).
- **High GPU utilization:** ORTHRUS achieves higher average GPU utilization by 40.6 percentage points, allowing it to complete a mixed workload, an average of 43% faster (Section 4.4).
- **Minimal Overhead:** ORTHRUS's combined batching of embedding and generation requests sustains nearly the same throughput as running on dedicated GPUs, with embedding throughput reduced only by the expected overhead of LoRA (Section 4.5).
- **Model Generalizability:** ORTHRUS works across different models and LoRA adapters; we validate it on 3 models with LoRA ranks up to 64, showing consistent improvements (Section 4.6).

## 4.1 BASELINE DEPLOYMENT SCENARIOS AND EVALUATION WORKLOADS

**Baselines** We compare ORTHRUS against several baselines. (**SplitGPU**) A deployment that runs separate models on separate GPUs. We denote the number of GPUs dedicated to embedding and generation as $X:Y$, written SplitGPU$_{X:Y}$. For example, SplitGPU$_{1:3}$ means one GPU serves embedding while three GPUs serve generation. (**SameGPU**) A setup where both embedding and generation requests share a single GPU but are processed sequentially without heterogeneous batching. In this case, requests alternate in a first-come, first-served manner (FCFS). (**Scheduling policies**) We consider two methods. FCFS batches requests across types without prioritization. IBS (ours) allocates slots proportionally to the number of embedding and generation requests in the queue.

**Workloads and Models** For ORTHRUS, embedding and generation requests are co-located on a GPU, using LoRA adapters for the embedding models. We use the Mistral 7B model (Jiang et al., 2023a) and the LoRA adapter for the e5-mistral-7b-instruct model (Wang et al., 2024) for our experiments. For SplitGPU deployments, e5-mistral-7b-instruct with the combined weights are used for embedding requests. We test with additional models in Section 4.6.

We generate synthetic workloads that simulate a mix of embedding and generation requests. Embedding requests consist of 128 tokens, while generation requests consist of a prompt of 128 tokens and a generation length of 512 tokens. We vary the ratio of embedding to generation requests to evaluate performance under different workload mixes.

**Hardware** We run experiments on a Slurm cluster with 4 NVIDIA A100 40GB GPUs. Each GPU runs a vLLM deployment (SplitGPU or ORTHRUS), and requests are distributed evenly across the 4 GPUs using round-robin scheduling. Clients issue requests asynchronously, so multiple requests may be in flight concurrently on each GPU.

**Throughput Metric** We report throughput as completed requests per second, normalizing embedding and generation requests to account for their different compute costs. Let $T_{\text{gen}}$ and $T_{\text{embed}}$ be the per-second request rates. Normalized throughput is $T_{\text{norm}} = 1.7 \cdot T_{\text{gen}} + T_{\text{embed}}$ where $1.7 \approx 21.9/12.9$ is the ratio of saturated embedding to generation throughput on our setup. This makes embedding and generation workloads comparable, following prior work on long decode vs. short prefill workloads (Zhong et al., 2024; Agrawal et al., 2024; Narayanan et al., 2020).

## 4.2 ORTHRUS IMPROVES THROUGHPUT

We evaluate the throughput of ORTHRUS across different ratios of requests, comparing against Split-GPU deployments and alternative schedulers. The experiment uses 512 clients, measured over 180 seconds after a 180-second warm-up. Fig. 3a reports normalized throughput. SplitGPU deployments underutilize GPUs when workload ratios diverge from the static split. For example, SplitGPU$_{1:3}$ and SplitGPU$_{3:1}$ both show significant drops when the workload skews away from their fixed allocation. SameGPU with FCFS improves utilization but suffers when workloads are balanced (e.g., 40–60% generation), as alternating schedules prevent full GPU usage. In contrast, ORTHRUS with FCFS and IBS achieves near-optimal throughput across all ratios. We discuss the limitations of FCFS in Section 4.3, and show how IBS avoids these issues while maintaining high throughput.

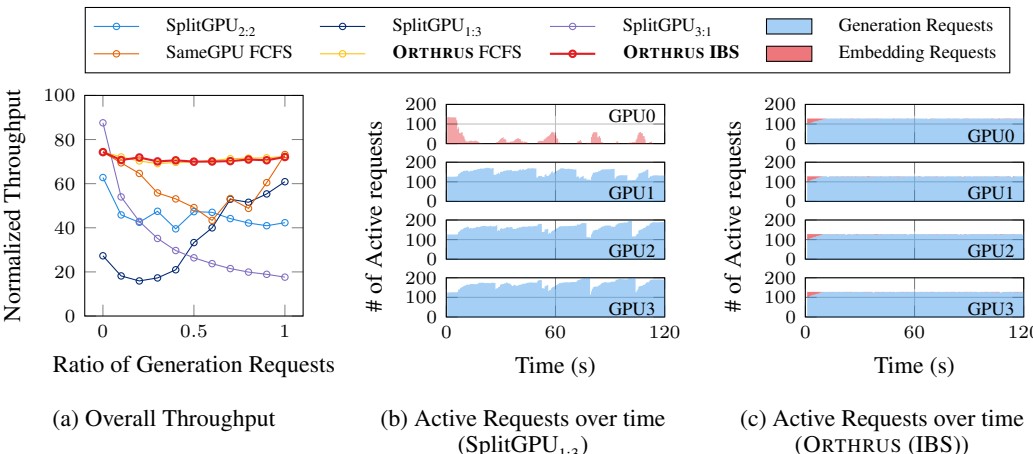

(a) Overall Throughput

(b) Active Requests over time (SplitGPU$_{1:3}$)

(c) Active Requests over time (ORTHRUS (IBS))

Figure 3: Throughput and GPU utilization comparison. (Fig. 3a) shows normalized throughput of existing serving techniques vs our heterogeneous batching, ORTHRUS. (Fig. 3b, Fig. 3c) shows request timelines for SplitGPU$_{1:3}$ and ORTHRUS under a 75% generation / 25% embedding workload. Each timeline reports the number of active requests per GPU. ORTHRUS fully utilizes all GPUs, while SplitGPU$_{1:3}$ leaves some underutilized.

Fig. 3b and Fig. 3c show GPU utilization under a workload of 75% generation and 25% embedding requests. In SplitGPU$_{1:3}$, the embedding GPU remains underutilized due to fixed model assignment, whereas ORTHRUS fully utilizes all GPUs with both request types. This highlights the limitation of static model placement: even when the GPU split (e.g., 1:3) matches the workload ratio, homogeneous batching leaves capacity unused due to variable decode lengths. As shown in Section 3.2, heterogeneous batching is inherently more efficient, as it can fill residual capacity in generation batches with embeddings, achieving higher throughput regardless of split accuracy. ORTHRUS adapts to workload dynamics and sustains high throughput across all ratios.

### 4.3 ORTHRUS WITH INTRA-BATCH SCHEDULING (IBS) REDUCES LATENCY

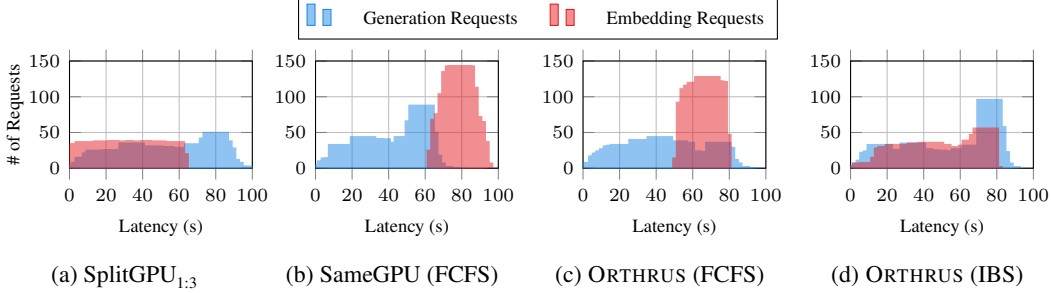

(a) SplitGPU$_{1:3}$

(b) SameGPU (FCFS)

(c) ORTHRUS (FCFS)

(d) ORTHRUS (IBS)

Figure 4: Latency distributions with various serving methods. Vertical bars appearing on the left side indicate lower latencies. The workload consists of 1,000 generations followed by 1,000 embeddings, all submitted at time 0. ORTHRUS with IBS achieves 9% lower p99 generation latency than SplitGPU, and up to 16% faster embedding latency than other schedulers.

We evaluate the latency distribution of embedding and generation requests under a workload where 1,000 generation requests are submitted immediately before 1,000 embedding requests. We measure per-request latency and plot the distributions in Fig. 4.

SplitGPU$_{1:3}$ allocates 3 GPUs to generation requests, and 1 GPU to embedding requests. Since requests are queued separately, embedding latency is unaffected by generation. However, once the 1,000 embedding requests are completed, the embedding GPU sits idle, while the generation model GPUs remain busy serving the remaining requests.

SameGPU deployment with FCFS achieves lower generation latency, because all 4 ORTHRUS instances can process generation requests. However, because SameGPU FCFS alternates between request types, embedding requests are forced to wait behind generation requests, producing a long tail in the embedding latency distribution. ORTHRUS deployment with FCFS improves upon SameGPU with FCFS, as it batches requests across types. As generation requests begin to finish, embedding requests are batched together and served. However, because FCFS does not distinguish between request types, embedding requests are still blocked behind a large number of generation requests, leading to a long tail in the embedding latency distribution.

ORTHRUS with IBS achieves low latency for both embedding and generation requests. The 99th percentile generation latency is reduced by 9% compared to SplitGPU ($89\,\mathrm{ms} \rightarrow 81\,\mathrm{ms}$). For embedding, IBS lowers the 99th percentile latency by 16% compared to other schedulers ($87\,\mathrm{ms} \rightarrow 73\,\mathrm{ms}$), while remaining within $16\,\mathrm{ms}$ of SplitGPU, which dedicates a GPU to embedding. These results show that IBS avoids embedding starvation while still reducing generation latency.

## 4.4 ORTHRUS ALLOWS HIGH GPU UTILIZATION WITH INSTANTANEOUS LOAD BALANCING

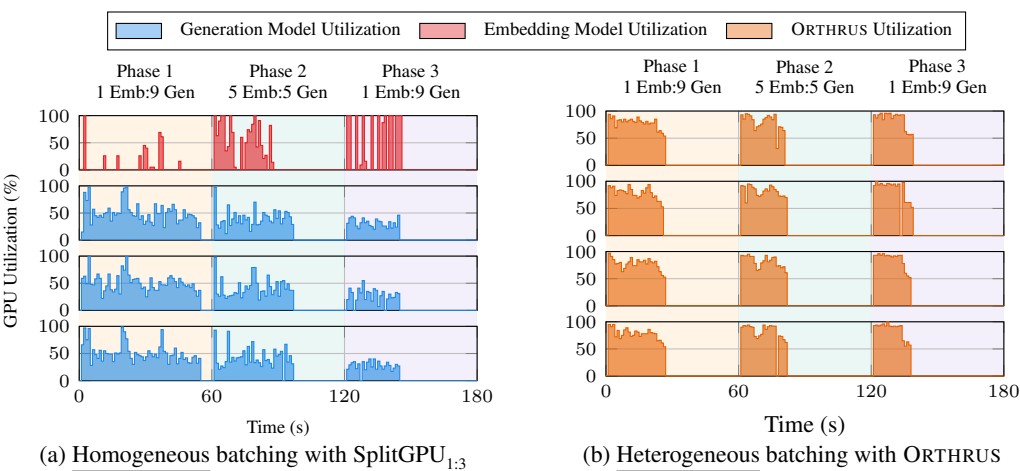

(a) Homogeneous batching with SplitGPU$_{1:3}$  (b) Heterogeneous batching with ORTHRUS

Figure 5: GPU utilization comparison between homogeneous batching (SplitGPU$_{1:3}$, Fig. 5a) and heterogeneous batching (ORTHRUS, Fig. 5b) under three phases of mixed workloads. In SplitGPU$_{1:3}$, the dedicated embedding GPU is severely underutilized in Phase 1, and generation GPUs in Phase 3, leading to lower average utilization (38%) and slower completion. In contrast, ORTHRUS keeps all GPUs uniformly busy across phases, sustaining much higher utilization (79%) and completing each phase 43% faster on average.

We evaluate the GPU utilization over different phases of request patterns to compare the load balancing capabilities of ORTHRUS, against a SplitGPU$_{1:3}$ deployment. The workload consists of three phases of 60 seconds each, with 1,000 requests per phase: Phase 1 has 10% embedding and 90% generation requests, Phase 2 has a 50/50 split, and Phase 3 has 90% embedding and 10% generation. Fig. 5 shows GPU utilization over time for both systems.

Overall, ORTHRUS completes each phase faster than SplitGPU, finishing requests an average 43% sooner. This improvement stems from higher GPU utilization: SplitGPU averages 38% during execution, whereas ORTHRUS reaches 79% (+40.6 pp). The gap is most pronounced in Phase 1, where the embedding GPU in SplitGPU is underutilized while ORTHRUS keeps all GPUs busy.

## 4.5 ORTHRUS SERVES BOTH REQUEST TYPES WITH MINIMAL OVERHEAD

We evaluate whether ORTHRUS can serve both embedding and generation requests on a single GPU without reducing throughput compared to dedicated deployments. We run embedding-only and generation-only workloads while varying concurrency from 1 to 256 clients. For embeddings (Fig. 6a), ORTHRUS achieves throughput close to a dedicated embedding model, with only a minor drop from LoRA overhead. For generation (Fig. 6b), ORTHRUS matches the dedicated generation

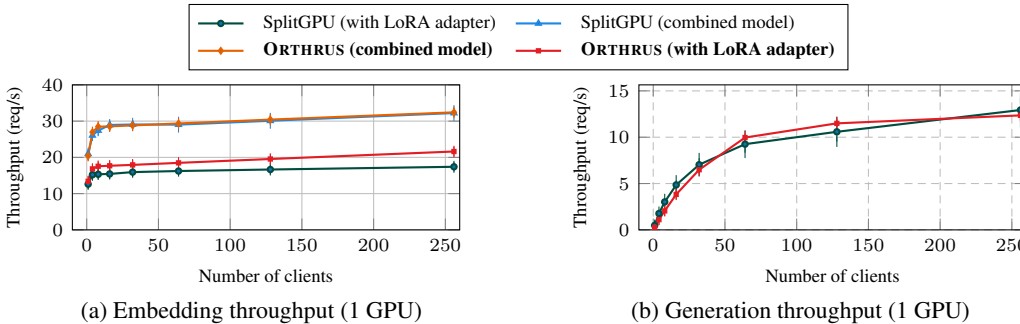

(a) Embedding throughput (1 GPU)     (b) Generation throughput (1 GPU)

Figure 6: Overhead analysis of ORTHRUS's heterogeneous batching. We plot throughput vs. number of clients on a single GPU for embedding-only (Fig. 6a) and generation-only (Fig. 6b) workloads. ORTHRUS achieves throughput comparable to dedicated vLLM SplitGPU deployments. Embedding throughput shows only a small drop due to LoRA overhead, while generation throughput matches SplitGPU across all concurrency levels.

deployment across all concurrency levels. Together, these results show that ORTHRUS preserves throughput for both request types while enabling them to share the same GPU.

## 4.6 ORTHRUS IS GENERALIZABLE TO OTHER MODELS

Table 1: Generalizability analysis by comparing throughput across different models and request mixes. We report embedding, generation, and the normalized throughput (Weighted Sum = Embed + $1.7 \times$ Gen) for three ratios (9:1, 5:5, 1:9; embedding:generation). SplitGPU baseline runs separate models on a 1:3 GPU split. ORTHRUS co-locates both tasks on every GPU.

| Model | Variant | 9 Emb:1 Gen | | | 5 Emb:5 Gen | | | 1 Emb:9 Gen | | |
|---|---|---|---|---|---|---|---|---|---|---|
| | | Embed | Gen | Weighted Sum | Embed | Gen | Weighted Sum | Embed | Gen | Weighted Sum |
| Mistral 7B | SplitGPU$_{1:3}$ | 15.64 | 1.49 | 18.02 | 12.54 | 11.97 | 32.89 | 5.10 | 29.56 | 55.35 |
| | ORTHRUS (e5-mistral LoRA) | 58.80 | 7.00 | **70.07** | 25.32 | 26.24 | **69.93** | 4.20 | 39.10 | **70.67** |
| Qwen2 7B | SplitGPU$_{1:3}$ | 13.23 | 1.72 | 16.15 | 10.72 | 10.08 | 27.86 | 2.81 | 27.76 | 50.00 |
| | ORTHRUS (LoRA Rank 32) | 40.90 | 4.40 | **48.38** | 26.80 | 25.93 | **70.88** | 6.20 | 48.23 | **88.19** |
| LLaMA3.1 8B | SplitGPU$_{1:3}$ | 13.43 | 1.32 | 15.67 | 10.18 | 9.95 | 27.10 | 3.06 | 25.63 | 46.63 |
| | ORTHRUS (LoRA Rank 64) | 56.46 | 5.90 | **66.49** | 34.60 | 37.50 | **98.35** | 4.23 | 47.32 | **84.674** |

Finally, we evaluate the generalizability of ORTHRUS across different models and LoRA ranks. We run experiments with Qwen2 7B (Yang et al., 2024) and LLaMA3.1 8B (Grattafiori et al., 2024), in addition to Mistral 7B, using synthesized LoRA adapters for embedding models. We vary the ratio of embedding to generation requests and report throughput for each configuration. Table 1 summarizes the results.

Across all models and request ratios, ORTHRUS consistently outperforms SplitGPU$_{1:3}$. Even under extreme skews (e.g., 9:1 or 1:9), ORTHRUS sustains high throughput. For example, with Mistral 7B at a 9:1 ratio, ORTHRUS achieves up to $3.8\times$ higher throughput than the baseline. These results demonstrate that ORTHRUS generalizes across different models and LoRA ranks, providing robust performance improvements over SplitGPU deployments. This suggests that ORTHRUS can be broadly applied to mixed embedding-generation workloads in diverse LLM serving scenarios.

## 5 CONCLUSION

This work presents ORTHRUS, a serving system that unifies embedding and generation within a unified model runner through heterogeneous batching. Unlike existing systems, ORTHRUS integrates embedding into the same scheduling and execution path as generation. With its new mechanisms, ORTHRUS achieves higher throughput and lower latency under mixed workloads, enabling efficient serving for modern knowledge-intensive LLM applications.

## REPRODUCIBILITY STATEMENT

We release anonymized source code and experiment scripts at `https://anonymous.4open.science/r/Orthrus-483B`. Our repository contains instructions for setting up the environment, launching experiments, and reproducing all figures and tables in the paper. The detailed evaluation setup is described in Section 4.1, and we provide configuration files for all baseline and ORTHRUS deployments. Together, these resources ensure that our results can be replicated and extended by the research community.

## ETHICS STATEMENT

This work does not involve human subjects, proprietary datasets, or personally identifiable information. Our contributions focus on system design for efficiently serving existing open-weight LLMs. Potential risks are those inherent to large language models, such as misuse for generating harmful content, which are unchanged by our methods. Finally, we used an LLM only for grammar checking in writing this paper.

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
