# OpenReview forum: "Efficient Embedding-Generation Serving with Heterogeneous Batching"
_ICLR.cc/2026/Conference — Submitted to ICLR 2026_

### Official Review · Reviewer_Ty67 · 2025-10-30

**Soundness:** 2
**Presentation:** 3
**Contribution:** 3
**Rating:** 6
**Confidence:** 3

**Summary:**

This Paper introduces a serving system that enables heterogeneous batching of embedding and generation requests within the same inference iteration. The key technical contribution is a unified runner abstraction, where both request types share the same schedule–forward–emit execution structure. To support this, the authors develop incremental pooling for embeddings, aligning their computation granularity with token-level decoding. Furthermore, an intra-batch scheduling policy dynamically adjusts batch composition to balance embedding and generation latency. Experimental results on multi-GPU clusters demonstrate that this new system achieves higher normalized throughput and substantially lower tail latency compared to static GPU splitting and homogeneous batching approaches, while maintaining compatibility with existing models and LoRA adapters.

**Strengths:**

- Theoretical Analysis: The work presents a clear theoretical model comparing homogeneous and heterogeneous batching, formally demonstrating how heterogeneous batching improves GPU utilization and reduces total inference steps.
- Implementation based on vLLM with Triton: The system is implemented by extending vLLM, retaining its asynchronous scheduling and PagedAttention memory management, ensuring practical deployability. Custom Triton kernels enable incremental pooling, allowing embedding computation to be aligned with token-level decoding.

**Weaknesses:**

- Insufficient background information for motivation: The paper does not quantitatively characterize the relative computational and memory costs of embedding versus generation workloads. Providing concrete measurements (e.g., FLOPs, bandwidth usage, KV-cache growth) would strengthen the motivation and clarify the severity of the imbalance the system aims to address.
- Only one primary baseline: The evaluation primarily compares against GPU-level model splitting, which limits the breadth and persuasiveness of the results. Including additional state-of-the-art serving systems or scheduling strategies would provide a more comprehensive and competitive baseline comparison.
- No scalability evaluation from single GPU to multi-GPU: The paper does not examine how the proposed method scales with increasing GPU count or model parallel configurations.
- Lack of discussion on broader inference task diversity: The work focuses solely on embedding and generative workloads, without considering other common inference tasks such as reranking, speculative decoding, or multi-modal encoders. Discussing how the proposed abstraction might generalize to these scenarios would improve completeness and applicability.

**Questions:**

- General questions are given in the weakness part.
- Will the experimental conclusions change when scaling from a single GPU to 2 / 4 / 8 GPUs?
- Is there a potential corner case where, under certain embedding–generation workload ratios, the baseline may outperform the proposed approach?‘
- Is the baseline operator implementation also based on Triton? Was there any comparison regarding operator-level performance? Is it possible that part of the overall performance improvement comes from optimizations at the operator implementation level rather than from the batching or scheduling design itself?
- In Table 1, as the ratio between the two task types varies, the baseline and the proposed framework exhibit different performance trends. Could this be further explained? For example, in the LLaMA3 row, the baseline performance increases steadily, whereas ORTHRUS increases first and then decreases.
- Have you experimented with different combinations of token lengths, and how would such variations affect the conclusions?

---

> ### Author Response · Authors · 2025-11-20
> **Response to Reviewer Ty67 (1/2)**
>
> We thank you for your helpful advice! We are happy to see you find our paper general and useful.
>
> ### (W2, W3, Q2, Q6) W2: disaggregated prefill-decoding;   W3,Q2: GPU scaling;  Q6: varying token lengths
>
> To address the concern, we have run new experiments as follows.
>
> **New Experiment**
>
> (W2) We have added a **vLLM-based disaggregated prefilling** baseline, which is a stronger baseline representative of state-of-the-art serving systems (TetriInfer, arXiv Jan 2024). Find the results in the tables below.
>
> (W3, Q2) We now include Iter-RetGen retrieval ([Shao et al., 2023](https://aclanthology.org/2023.findings-emnlp.620/)) running on the 2WikiMultihopQA benchmark dataset, **on 1, 2, and 4 GPUs**. In this setup, embedding and generation alternate, which are then followed by a final long generation step.
>
>
> (Q6) Our new 2WikiMultihopQA RAG evaluation has variable-length prompts (avg 500 tokens, max 2000 tokens), variable and long decode lengths (avg 3000 tokens, max 4000 tokens), multi-document retrieval (1-10 documents), and re-embedding phases.
>
> Our system (Orthrus) improves query throughput, decode-token throughput, and end-to-end latency.  These results demonstrate that Orthrus remains effective under (more) realistic, heterogeneous input-length and output-length distributions. We will include these as primary experiments in the revised manuscript.
>
> - End-to-end p99 latency (ms)
>
> |                           | 1GPU | 2GPU | 4GPU       |
> | ------------------------- | ---- | ---- | ---------- |
> | SplitGPU                  | N/A |960078.24| 1342024.58 |
> | SameGPU (homo batching)   |673403.17|675996.76| 700557.74  |
> | Disaggregated vLLM        | N/A |602607.18| 617047.52  |
> | **Orthrus (ours)**        |**573926.67**|**600510.98**| **593397.97**  |
>
>
> - Embedding throughput (requests/sec)
>
> |                           | 1GPU | 2GPU | 4GPU  |
> | ------------------------- | ---- | ---- | ----- |
> | SplitGPU                  | N/A  |0.08  | 0.119 |
> | SameGPU (homo batching)   | 0.067| 0.126| 0.252 |
> | Disaggregated vLLM        | N/A  |0.118 | 0.276 |
> | **Orthrus (ours)**        | **0.075**|**0.142** | **0.288** |
>
> - Generation throughput (requests/sec)
>
> |                         | 1GPU | 2GPU | 4GPU  |
> | ----------------------- | ---- | ---- | ----- |
> | SplitGPU                | N/A  |0.11  | 0.141 |
> | SameGPU (homo batching) |0.08  |0.148 | 0.296 |
> | Disaggregated vLLM      | N/A  |0.14  | 0.327 |
> | **Orthrus (ours)**      |**0.088** |**0.166** | **0.336** |
>
>
> ### (Q3) Is there a corner case where baselines outperform Orthrus?
>
> Orthrus does introduce some memory overhead due to the inclusion of additional LoRA weights. However, as demonstrated in Figure 6 of the manuscript, the only measurable overhead incurred by heterogeneous batching arises from these LoRA weights. In all other respects, Orthrus’s performance aligns closely with the baseline. In corner cases, such as extensive decoding or complete KV cache saturation, Orthrus naturally reverts to pure-generation iterations that are functionally identical to the baseline, with no additional penalty.
>
>
> ### (Q4) Is operator-level implementation responsible for the improvements?
>
> No. All baselines (SameGPU, SplitGPU) and Orthrus use the same vLLM operator suite, including FlashAttention, KV-cache, sampling, LoRA application, and prefill and decode kernels.
>
> Orthrus adds only a lightweight incremental-pooling kernel (sub-millisecond), which is not on the decoding critical path. Therefore, improvements arise purely from batch composition and memory-opportunistic scheduling, not from kernel differences.

---

> ### Author Response · Authors · 2025-11-20
> **Response to Reviewer Ty67 (2/2)**
>
> ### (Q5) Why do Table 1 and Figure 5 show different throughput trends?
>
> This discrepancy comes from the normalized throughput metric, whose weighting constant was measured only for Mistral-7B under a specific 128-token embedding and ~128→512-token generation workload. When applied to different models or prompt-length, the normalized axis no longer maps linearly to absolute performance.
>
> In response, we will report absolute throughput and latency separately for embedding and generation.
>
> **Revised table 1 with absolute throughput**
>
> | Model        | Variant                     | 9:1 (Emb) | 9:1 (Gen) | 5:5 (Emb) | 5:5 (Gen) | 1:9 (Emb) | 1:9 (Gen) |
> |--------------|-----------------------------|-----------|-----------|-----------|-----------|-----------|-----------|
> | Mistral 7B   | SplitGPU₁:₃                 | 15.64     | 1.49      | 12.54     | 11.97     | 5.10      | 29.56     |
> |              | **Orthrus** (e5-mistral LoRA)   | 58.80     | 7.00      | 25.32     | 26.24     | 4.20      | 39.10     |
> | Qwen2 7B     | SplitGPU₁:₃                 | 13.23     | 1.72      | 10.72     | 10.08     | 2.81      | 27.76     |
> |              | **Orthrus** (LoRA Rank 32)      | 40.90     | 4.40      | 26.80     | 25.93     | 6.20      | 48.23     |
> | LLaMA3.1 8B  | SplitGPU₁:₃                 | 13.43     | 1.32      | 10.18     | 9.95      | 3.06      | 25.63     |
> |              | **Orthrus** (LoRA Rank 64)      | 56.46     | 5.90      | 34.60     | 37.50     | 4.23      | 47.32     |
>
>
>
> ### (W1) Insufficient background on computational and memory imbalance
>
> To address this concern, we have conducted a micro benchmark running pure embedding and generation workloads. As shown in the table below, embedding operations achieve high mean GPU Utilization (85.3%) with low memory usage (36.5%), whereas generation operations show high GPU memory utilization (90.5%) with a lower mean GPU compute utilization (75.5%).
>
>
> - GPU Utilization
>
> |            | Mean | Max |
> |------------|------|-----|
> | Embedding (compute-hungry) |**85.3%** |98.0%|
> | Generation |**75.5%** |98.0% |
>
>
> - GPU Memory Utilization
>
> |            | Mean | Max |
> |------------|------|-----|
> | Embedding  |**36.5%** |37.1%|
> | Generation (memory-hungry) |**90.5%** |90.5%|
>
> These results demonstrate that embedding is compute bound while generation is memory bound. ***By combining those workloads within the same kernel***, our system achieves higher overall GPU utilization, effectively leveraging both compute and memory resources. No prior work demonstrated this benefit of ***intra-kernel heterogenous batching***.
>
>
>
> ### (W4) Lack of generalization beyond embedding/generation
>
>
>
> Our abstraction directly generalizes to any prefill-only task, including reranking, multi-modal encoders, or any single token generation tasks. These tasks share a common computational structure, as follows.
>
> Given an input sequence $X=(x_1,\dots,x_n)$, the model produces hidden states $H=(h_1,\dots,h_n)$ in one forward pass, and the task’s output is obtained by applying some function $g$ to these states: $y = g(H)$.  Here $g$ represents a task-specific aggregator or head (mean pooling across $H$ for embedding, a classification/scoring head for reranking, or a next-token probability distribution for single-token generation), and crucially $g$ does not depend on any further model steps beyond the prompt.
>
> Embedding is a specific instance of this framework: for an embedding request, $g$ can be instantiated as a pooling operator (e.g. $g(H) = \frac{1}{n}\sum_{t=1}^n h_t$ for mean pooling) that computes the final vector from the forward-pass hidden states
>
> The unified kernel executes the identical transformer forward pass for every request type (differing only in input length) and then performs the appropriate “emit” operation for each request (e.g. sampling for generation or pooling for an encoding task) in parallel. Thus, any inference that completes in one forward pass falls under our abstraction and can be seamlessly batched and executed alongside standard generation tasks, confirming the generality of our approach.

---

### Official Review · Reviewer_7SYe · 2025-10-31

**Soundness:** 3
**Presentation:** 3
**Contribution:** 3
**Rating:** 4
**Confidence:** 4

**Summary:**

This paper mixes embedding requests and generation requests in the same batch, achieving higher throughput and lower latency.

**Strengths:**

* Solid implementation.
* Important problem and good results.

**Weaknesses:**

* Not sure about the validity behind the assumptions.

**Questions:**

* Are you assuming that embedding models and the generation LLMs are the same model or not? If this is the case, the split GPU solution  can already work. If this is not the case, correct me if I am wrong, it means that your system also needs to store both the embedding models and the generation LLMs into GPU. SplitGPU can do the same thing to dynamically change the ratio between embedding LLMs and generation LLMs without reloading the model.
* The evaluation assumes the embedding LLMs and generation LLMs are LoRA-finetuned from the same base model. This assumption might be too strong: for example, Qwen embedding and Qwen reranking LLMs are two different models. And the size of embedding and reranking models may also be different. Will this change your evaluation takeaway?
* What are the use cases where the # of generation requests : # of embedding requests significantly vary over time? For example, in recommendation system, the # of generation requests : # of embedding requests is fixed because for each query it typically retrieves the same amount of related documents for recommendation.
* The LLMs are too large for information retrieval at scale. Typically for retrieval the LLMs are <1B. Will this change your evaluation takeaway or not?

---

> ### Author Response · Authors · 2025-11-20
> **Response to Reviewer 7SYe (1/2)**
>
> Thanks for your careful review. We are glad that you like our paper and find our results solid. We answer your questions below:
>
> ### (W1, Q1) Are you assuming the embedding and generation models are the same model? If so, doesn’t SplitGPU already solve this? If not, wouldn’t your method need to store both models on the GPU?
>
> **Clarification**
> We don't assume the same (base) model. Our core technical contribution lies in **heterogenous batching**. Specifically, there are two closely related, but orthogonal aspects: (1) memory efficiency and (2) heterogeneous batching. The first is memory efficiency through LoRA adapters or consolidated models, which benefits our approach; however, our approach still works even when two models are completely separate. The second is heterogeneous batching, i.e., the ability to schedule both embedding and generation requests within the same kernel.
>
>
> **New Experiment**
> To demonstrate that our heterogeneous batching is orthogonal to parameter sharing, we have designed a new experiment. We run two full models on intra-batch scheduling by running two Qwen 0.6B models with Orthrus. We compare this to a baseline where we deploy two vLLM instances on one GPU, each running Qwen 0.6B embedding model and Qwen 0.6B generation model. We run Iter-RetGen retrieval ([Shao et al., 2023](https://aclanthology.org/2023.findings-emnlp.620/)) on the 2WikiMultihopQA benchmark dataset. Under equivalent memory pressure (i.e., by ensuring all methods have the same memory besides model weights), Orthrus achieves higher throughput and lower latency.
>
> - Throughput (request/sec): the higher the better
>
> |                    | Embedding request throughput | Generation request throughput | Query Throughput |
> |--------------------|------------------------------|-------------------------------|------------------|
> | Two distinct kernels        |0.214                         |0.248                          |0.036             |
> | Kernels w/ homogenous batching    |0.184       |0.214                          |0.03              |
> | **Orthrus (ours): Kernels w/ heterogeneous batching** | **0.265** | **0.308**            | **0.043**             |
>
> - p99 Latency (ms): the lower the better
>
> |                    | Embedding request p99 latency | Generation request p99 latency| Query e2e  p99 latency|
> |--------------------|-------------------------------|-------------------------------|-----------------------|
> | Two distinct kernels        |85.57                          |79301.89                       |568170.36              |
> | Kernels w/ homogenous batching    |51784.73*    |65910.65                       |653591.01              |
> | **Orthrus (ours): Kernels w/ heterogeneous batching** | **53.19** | **29855.44** |**228596.96** |
>
>
> $*$ High embedding latency arises because the FCFS scheduler forces embedding requests to wait behind long-running generation requests, as shown in Fig. 4.
>
>
> ### (Q2) The evaluation assumes embedding and generation are LoRA-finetuned from the same base model. Is this assumption too strong? What about cases like Qwen embedding vs. Qwen reranking (different models)?
>
> **Clarification**
> Our idea of heterogeneous batching of embedding and generation is not limited to LoRA finetuned models. Specifically, our approach is applicable to any combination of embedding and generation models if a kernel can access those weights. Unfortunately, vLLM's current limitation — one model for one runner — makes full evaluation challenging. Nevertheless, we have conducted a new experiment where two independent models are loaded into one GPU; Orthrus still outperforms two separate deployments of embedding and generation models.
>
> **New Experiment**
> Please see the tables under **Q1**.

---

> ### Author Response · Authors · 2025-11-20
> **Response to Reviewer 7SYe (2/2)**
>
> ### (Q3) What use cases have highly variable embedding:generation ratios? Many recommendation systems have a fixed ratio.
>
> Many modern LLM workflows exhibit variations in embedding and generation ratios.
>
> - Embedding-heavy phases
>   - The LlamaIndex SubQuestionQueryEngine workflow breaks a complex query into multiple sub-questions, each triggering its own retrieval and embedding step - Retrieval query type induces different retrieval fan-outs, requiring multiple embedding per query. ([SubQ](developers.llamaindex.ai/python/examples/query_engine/sub_question_query_engine/))
>   - Upgrading embedding inside vector DB. Large-scale vector-DB maintenance (refreshing or re-embedding documents).
>
> - Generation-heavy phases
>   - Agentic reasoning loops (e.g. [Wu et al., ACL 2025](https://aclanthology.org/2025.acl-long.1383/)) using an LLM + tool-agent loop to generate multiple potential answers from one retrieved context, generating multiple potential answers.
>   - Conversational LLM applications skipping retrieval after the first round of retrieval.
>
> In order to test our system for a full spectrum of scenarios, we varied the raio from one end to the other end.
>
> **New Experiment**
> In addition to the existing controlled experiments, we now include Iter-RetGen retrieval ([Shao et al., 2023](https://aclanthology.org/2023.findings-emnlp.620/)) running on the 2WikiMultihopQA benchmark dataset. Specifically, embedding and generation alternates, which are then followed by a final long generation step. We ran 100 queries, each randomly retrieving between 1 to 10 documents. This totaled 580 embedding requests, and 680 generation requests in total, showing a similar 1:1 ratio over the experiment.
>
> The evaluation results are shown below. In this (more) realistic workload, our approach (Orthrus) achieves both faster latency and higher throughput.
>
> **End-to-end per-request p99 latency (ms)**
> |                           | 1GPU | 2GPU | 4GPU       |
> | ------------------------- | ---- | ---- | ---------- |
> | SplitGPU                  | N/A |960078.24| 1342024.58 |
> | SameGPU (homo batching)   |673403.17|675996.76| 700557.74  |
> | Disaggregated vLLM        | N/A |602607.18| 617047.52  |
> | **Orthrus (ours)**        |**573926.67**|**600510.98**| **593397.97**  |
>
> **Embedding throughput (requests/sec)**
> |                           | 1GPU | 2GPU | 4GPU  |
> | ------------------------- | ---- | ---- | ----- |
> | SplitGPU                  | N/A  |0.08  | 0.119 |
> | SameGPU (homo batching)   | 0.067| 0.126| 0.252 |
> | Disaggregated vLLM        | N/A  |0.118 | 0.276 |
> | **Orthrus (ours)**        | **0.075**|**0.142** | **0.288** |
>
> **Generation throughput (requests/sec)**
> |                         | 1GPU | 2GPU | 4GPU  |
> | ----------------------- | ---- | ---- | ----- |
> | SplitGPU                | N/A  |0.11  | 0.141 |
> | SameGPU (homo batching) |0.08  |0.148 | 0.296 |
> | Disaggregated vLLM      | N/A  |0.14  | 0.327 |
> | **Orthrus (ours)**      |**0.088** |**0.166** | **0.336** |
>
>
>
>
> ### (Q4) LLMs are too large for retrieval at scale; typically embedding models are <1B. Does this invalidate the evaluation?
>
> **Clarification**
> Recent empirical studies show large embedding models (e.g., 7B-scale) significantly improve retrieval quality, multi-hop reasoning, and reranking accuracy compared to sub-1B models ([Wang et al., ACL 2024](https://aclanthology.org/2024.acl-long.642/), [Behnamghader et al., COLM 2024](https://arxiv.org/abs/2404.05961)).
>
> **New Experiment**
> While our primary focus is relatively large embedding models (e.g., Qwen3-Embedding-8B, e5-mistral-7b-instruct), we have added a new experiment using smaller models (Qwen3-0.6B-Instruct and Qwen3-Embedding-0.6B). The detailed results are shown under **(Q1)**. Our approach achieved the highest performance.

---

### Official Review · Reviewer_DVbn · 2025-11-01

**Soundness:** 2
**Presentation:** 3
**Contribution:** 2
**Rating:** 2
**Confidence:** 4

**Summary:**

Existing retrieval augmented generation systems suffer from low GPU utilization due to the inefficiency of mixed workloads consisting of embedding and generation requests, which reduces overall throughput and leads to high latency. To mitigate this issue, the authors propose ORTHRUS, a framework that enables heterogeneous batching. Through a unified kernel abstraction and fine-grained intra-batch scheduling, embedding and generation requests can be processed within the same batch, thereby improving hardware utilization. Experimental results demonstrate that the proposed method achieves both higher throughput and lower latency across various setups.

**Strengths:**

1. The motivation to improve mixed workloads in RAG systems is clear, practical, and reasonable.

2. The proposed idea is interesting and has potential.

3. Overall, the paper is well-structured and easy to follow.

**Weaknesses:**

1. The evaluation workload does not align with real-world scenarios:

   a. Dependency between embedding and generation requests is ignored: In RAG, embedding requests typically come before generation requests. However, the evaluation assumes that 1000 generation requests are issued before 1000 embedding requests, which is impractical for typical RAG workloads.

   b. Input length differences are ignored: Retrieved information is usually in the form of paragraphs or articles, making generation inputs much longer than embedding queries. Yet, in the experiments, both request types use an identical input length of 128 tokens, which is not representative of real-world scenarios.

   c. The embedding-to-generation ratio is impractical: Generally, each generation process involves one or no embedding request, so the number of generation requests should be greater than the number of embedding requests. If multiple embeddings are used in one generation, the generation inputs become significantly longer, and the evaluation should be adjusted accordingly.

2. Normalized throughput is not a common or fair metric:

   The throughput of embedding and generation requests heavily depends on input length, model, and hardware. Moreover, GPU utilization is not saturated in most cases (as shown in Figure 5). Thus, applying a constant to compute a weighted sum is unjustified. This metric also lacks interpretability and comparability across different settings. Reporting the overall time and separately showing the latency and throughput for embedding and generation requests would be more appropriate.

3. The baseline comparison might not be fair:

   In the SplitGPU method, the GPU becomes idle after completing all embedding requests. A simple optimization that allows the GPU to process generation requests afterward would significantly improve the baseline performance.

4. Results are self-contradictory:

   Figure 5 and Table 1 lead to opposite conclusions. In Figure 5, the throughput increases more significantly with a higher ratio of generation requests, while Table 1 shows the opposite trend. Additionally, Figure 5 contains a labeling error: Phase 3 should be [9 Emb : 1 Gen] rather than [1 Emb : 9 Gen] as stated in the text, which causes major confusion.

5. Limited evaluation scope:

   Experiments are conducted only on a single system (4 × A100) with one request type (128-token input). This setup does not sufficiently support the generalizability of the results to diverse workloads in real-world scenarios.

**Questions:**

1. Regarding workloads:

   a. In what scenario would 1000 generation requests occur before 1000 embedding requests?

   b. In what case would generation and embedding requests have the same input length (128 tokens)?

   c. Under what conditions would the ratio of embedding to generation requests be as extreme as 1:9 or 9:1?

2. How does the constant for normalized throughput vary with different input lengths, model types, or hardware configurations?

3. Does the proposed method perform better under embedding-heavy workloads or generation-heavy workloads?

4. Does the method rely on LoRA-based embedding models? How would it perform if the embedding and generation models are entirely different?

5. How could SplitGPU and the proposed method be adapted for a single-GPU setup (as in Figure 6)?

6. Could the authors evaluate the framework on a practical RAG system and report improvements over the original implementation?

---

> ### Author Response · Authors · 2025-11-20
> **Response to Reviewer DVbn (1/2)**
>
> Thanks for your careful review. We are glad that you find our research interesting and relevant. We answer your questions below.
>
>
> ### (W1, W5, Q1, Q8) Workload Clarification and Improvement
>
> #### (Q1.a) In what scenario would 1000 generation requests occur before 1000 embedding requests?
>
>
> **Clarification**
> Existing methods (e.g., Iter-RetGen [(Shao et al., 2023)](https://aclanthology.org/2023.findings-emnlp.620/), Self-RAG [(Asari et al., ICLR 2024)](https://openreview.net/forum?id=hSyW5go0v8), FLARE [( Jiang et al., EMNLP 2023)](https://aclanthology.org/2023.emnlp-main.495/), HyDE [(Gao et al., ACL 2023)](https://aclanthology.org/2023.acl-long.99/)) perform a series of (embedding) - (generation) - (embedding) - (generation) and so on, where our evaluation isolates an intermediate [(generation) - (embedding)] pair to specifically test potential long waits.
>
>
>
> **New Experiment**
> To address the concern, we now include Iter-RetGen retrieval ([Shao et al., 2023](https://aclanthology.org/2023.findings-emnlp.620/)) running on the 2WikiMultihopQA benchmark dataset. Specifically, embedding and generation alternates, which are then followed by a final long generation step. The evaluation results are shown below. In this (more) realistic workload, our approach (Orthrus) achieves both faster latency and higher throughput compared to existing systems (including prefill/decoding disaggregation newly pointed by Reviewer1).
>
> - End-to-end p99 latency (ms)
>
> |                           | 1GPU | 2GPU | 4GPU       |
> | ------------------------- | ---- | ---- | ---------- |
> | SplitGPU                  | N/A |960078.24| 1342024.58 |
> | SameGPU (homo batching)   |673403.17|675996.76| 700557.74  |
> | Disaggregated vLLM        | N/A |602607.18| 617047.52  |
> | **Orthrus (ours)**        |**573926.67**|**600510.98**| **593397.97**  |
>
> - Embedding throughput (requests/sec)
>
> |                           | 1GPU | 2GPU | 4GPU  |
> | ------------------------- | ---- | ---- | ----- |
> | SplitGPU                  | N/A  |0.08  | 0.119 |
> | SameGPU (homo batching)   | 0.067| 0.126| 0.252 |
> | Disaggregated vLLM        | N/A  |0.118 | 0.276 |
> | **Orthrus (ours)**        | **0.075**|**0.142** | **0.288** |
>
> - Generation throughput (requests/sec)
>
> |                         | 1GPU | 2GPU | 4GPU  |
> | ----------------------- | ---- | ---- | ----- |
> | SplitGPU                | N/A  |0.11  | 0.141 |
> | SameGPU (homo batching) |0.08  |0.148 | 0.296 |
> | Disaggregated vLLM      | N/A  |0.14  | 0.327 |
> | **Orthrus (ours)**      |**0.088** |**0.166** | **0.336** |
>
>
> #### (W1, Q1.b) In what case would generation and embedding requests have the same input length (128 tokens)?
>
> **Clarification**
> Unlikely in real-world workloads. Our evaluation used a controlled setting to isolate scheduling effects and uncertain generation lengths without confounding variability.
>
>
> **New Experiment**
> We have added a new RAG experiment with variable-length prompts (avg 500 tokens, max 2000 tokens), variable and long decode lengths (avg 3000 tokens, max 4000 tokens), multi-document retrieval (1-10 documents), and re-embedding phases. We ran 100 queries, each randomly retrieving between 1 to 10 documents. This totaled 580 embedding requests, and 680 generation requests in total. The results are shown in the above tables (under **Q1.a**), where our approach achieve the highest performance (i.e.,  lower latency and higher throughput).
>
>
> #### (Q1.c) Under what conditions would the ratio of embedding to generation requests be as extreme as 1:9 or 9:1?
>
> Such skews can occur when filling vector databases with new documents (embedding dominant workloads) or when multiple RAG pipelines reuse cached embeddings and mostly issue generation requests (generation dominant workloads).
>
> The experiment itself is not meant to represent a typical RAG trace, but a controlled stress test to demonstrate robustness under diverse ratio ratios. Our new experiment results (repoted under **Q1.a**) reflect a more typical setting (580 embedding requests to 680 generation requests).
>
> ### (Q2, W2) How does the constant for normalized throughput vary across models, input lengths, or hardware?
>
> **Clarification**
> The constant is empirical and reflects the steady-state throughput ratio between embedding and generation for a specific model and prompt length. It naturally varies with model type, prompt/decode length, and GPU hardware. It is not used in any scheduling decisions.
>
> In response to the reviewer’s concern, we now additionally report absolute throughput and latency for each request type, eliminating reliance on the normalized metric.

---

> ### Author Response · Authors · 2025-11-20
> **Response to Reviewer DVbn (2/2)**
>
> ### (Q3) Does Orthrus perform better under embedding-heavy or generation-heavy workloads?
>
> Orthrus performs well in both. The scheduler opportunistically fills unused GPU capacity with whatever request type is available. When only one request type is present, Orthrus behaves like a pure generation or pure embedding server. When both appear, Orthrus interleaves them to improve utilization.
>
>
>
> ### (Q4) Does the method rely on LoRA-based embedding models? How would it perform if the embedding and generation models are entirely different?
>
>
> **Clarification**
> Orthrus relies on LoRA to share base model weights between two different model. If embedding and generation models are completely disjoint, our method can still unify batching as long as both models fit in memory simultaneously. In such cases the performance gains would be smaller due to the absence of shared weights. We will clarify this limitation in the final version.
>
>
>
> ### (Q5) How could SplitGPU and Orthrus be adapted for a single-GPU setup?
>
> **Clarification**
> SplitGPU on a single GPU would require repeatedly loading and unloading full models (e.g., 8B params), which can take long. SameGPU already reflects the improved variant suggested by the reviewer, allowing both request types on one GPU without reloading. Orthrus naturally extends to the single-GPU case by interleaving embedding and generation within the same iteration. The results with different numbers of GPUs (1, 2, and 4) all reported under **Q1.a** together.

---

### Official Review · Reviewer_PhM6 · 2025-11-02

**Soundness:** 3
**Presentation:** 3
**Contribution:** 2
**Rating:** 4
**Confidence:** 3

**Summary:**

This paper targets a practical serving gap: current LLM serving systems handle embeddings and text generation as different workloads, which causes low GPU utilization when the incoming mix drifts. The authors introduce a unified runner that makes embeddings follow the same iteration-level control flow as decoding, enabled by an incremental pooling kernel. On top of this, they add an intra-batch scheduler that matches the number of embeddings vs. generations in a batch to the current queue sizes. On 4×A100 they show 1.28–4.52× higher normalized throughput and lower p99 latency than GPU-split or naïve shared-GPU baselines. The idea is neat and the implementation on vLLM is likely to be useful for practitioners. However, some of the novelty overlaps with or is at least very close to contemporaneous systems that already pack heterogeneous workloads (Sarathi-Serve, TetriInfer/ShuffleInfer, MACE, SageServe), and the experimental section does not compare against these, so the strength of the claim “first to serve embedding+generation in one batch” is weaker than written.

**Strengths:**

- Clear problem, tight abstraction. The unified iteration that ends with either sampling or incremental pooling is simple and well motivated; it’s the minimum change that makes embeddings schedulable alongside decodes.

- End-to-end implementation on a real engine. Building on vLLM (0.92) rather than a toy server makes this paper much more credible for ICLR systems readers. The design is compatible with paged KV and LoRA pinning.

- Solid throughput numbers on skewed mixes. The paper convincingly shows that any static GPU partition will be bad under changing embed/gen ratios, and ORTHRUS does fix exactly that.

- Reasonable latency story. The IBS derivation is standard queue-proportional batching; the fact that it tracks the empirical p99s is nice.

- Timeliness. vLLM’s own FAQ still says embeddings don’t benefit from its batching pipeline; the paper addresses that missing piece.

**Weaknesses:**

- Positioning vs. very recent work is incomplete. Disaggregated mixed-workload serving (TetriInfer, ShuffleInfer), memory-aware heterogeneous binning (MACE, SageServe), and even 2025 vLLM deployments with disaggregated P/D all address the same underlying challenge — simultaneous, heterogeneous requests — but the paper does not implement or report against them. This makes the "4.52×" headline look mostly like "we compared to a weak static-split baseline."

- Synthetic and narrow workloads. All evaluations use fixed-length synthetic traces, one LoRA per GPU, and a single model family; modern RAG pipelines and multi-agent apps have much broader length and adapter distributions. This matters because the key benefit of the unified runner is packing the residual memory — which is exactly what gets harder under wide length variance.

- The theory is nice but optimistic. The analytical model assumes generation cost scales nicely as sg=F·se and memory as mg=F·me, and that embedding requests can always fit in the "r" slots. In practice, stepping through decode with a very long output, or serving multiple LoRAs, can break this neat proportionality. The paper does not show a sensitivity analysis to such violations.

- Missing system details. The paper says it "invokes sampler and pooler kernels in parallel" but doesn’t give kernel-level timings or overheads; that matters because engines like Punica and S-LoRA have already shown that kernel-fusion/batching for heterogeneous LoRA adapters is doable with ~2 ms overhead, so ORTHRUS should benchmark against that bar.
proceedings.mlsys.org

- No real-trace or application-level eval. There is no experiment on an actual RAG loop (embed → retrieve → generate → re-embed). That would be the cleanest place to demonstrate end-to-end latency/throughput improvement.

- Novelty claim should be narrowed. Because 2025 systems like MACE actually do memory-aware batching of unrelated tasks in the same iteration, and SageServe does holistic scheduling across SLA tiers, the paper should claim "first to do embedding-aware heterogeneous batching with an incremental pooling kernel inside vLLM" — which is still good, just narrower.

**Questions:**

- Positioning against recent mixed-workload serving systems.
Sarathi-Serve, TetriInfer (“Inference without Interference”), and newer memory-aware schedulers like MACE all already batch or co-schedule heterogeneous requests at the iteration level. What, concretely, can your “unified runner + incremental pooling” do that these systems cannot? Please give a side-by-side capability table.

- Missing strong baselines.
You mainly compare to static GPU splits / naïve shared-GPU. Why didn’t you evaluate against (i) Sarathi-Serve configured for mixed prefills, or (ii) TetriInfer-style disaggregated executors, or (iii) a memory-aware bin-packing baseline like MACE? Do you expect your 4.5× speedup to still hold under those?

- Robustness of the “fill residual with embeddings” assumption.
Your analysis assumes leftover memory slots can almost always be filled by embeddings. How does the method behave when decodes are long, KV is fragmented, or multiple LoRAs/models are active so that no embedding fits — the exact cases that TetriInfer/MACE warn about? Please provide a sensitivity or failure-mode experiment.

---

> ### Author Response · Authors · 2025-11-20
> **Response to Reviewer PhM6 (1/2)**
>
> We thank the reviewer for the detailed and constructive comments.  Below, we address each concern in order.
>
> ### (W1, Q1, Q2) Missing positioning and empirical comparison against recent mixed-workload systems (e.g., TetriInfer, ShuffleInfer, MACE, SageServe, Sarathi-Serve)
>
> **Novelty**
> Our intra-kernel heterogenous batching is orthogonal to or different from all existing work.
> - Sarathi-Serve's chunked-prefill is already integrated into vLLM and is therefore part of all our baselines, which we will clarify.
> - Diaggregated inference (i.e., TetriInfer (arXiv Jan 2024) and ShuffleInfer) addresses "contention and head-of-line blocking" by splitting prefilling/decoding onto *dedicated GPUs*. In contrast, we show that the issue can be addressed ***even within each GPU*** using ***intra-kernel heterogeneous batching***, which is empirically more efficient.
> - MACE (arXiv Sep 2025) serves inference and fine-tuning using *two completely different systems* (vLLM and NotReported). In contrast, we demonstrate heterogenous requests can be scheduled ***within a single vLLM kernel***, achieving production-scale efficiency.
>
> We provide a comparison table below (Q1).
>
> | Capability              | Sarathi-Serve      | TetriInfer | MACE | Orthrus |
> | ---------------------------------- | :--------------------: | :--------: | :--: | :-----: |
> | Embedding-aware batching       |           x            |     x      |  x   |    o    |
> | Incremental pooling inside iteration |           x            |     x      |  x   |    o    |
> | Unified iteration for embed + decode | Partial (prefill only) |     x      |  x   |    o    |
> |  Opportunistic residual-memory fill  |           x            |     o      |  o   |    o    |
>
>
> **New Experiments**
> We have newly implemented ***disaggregated prefilling with vLLM*** baseline (i.e., the core of TetriInfer and ShuffleInfer) and have run additional experiments using 1 / 2 / 4 GPUs.  *Implementation Details:* While disaggregated serving and memory-aware heterogeneous binning lack open-source implementation (i.e., TetriInfer, ShuffleInfer, MACE) or are limited to simulation without actual LLM kernels (i.e., SageServe), we could timely implemented their core mechanisms inside vLLM. We present overall results first and then describe evaluation setup in detail.
>
> - End-to-end p99 latency (ms): the lower the better
>
> |                           | 1GPU | 2GPU | 4GPU       |
> | ------------------------- | ---- | ---- | ---------- |
> | SplitGPU                  | N/A |960078.24| 1342024.58 |
> | SameGPU (homo batching)   |673403.17|675996.76| 700557.74  |
> | Disaggregated vLLM        | N/A |602607.18| 617047.52  |
> | **Orthrus (ours)**        |**573926.67**|**600510.98**| **593397.97**  |
>
>
> - Embedding throughput (requests/sec): the higher the beter
>
> |                           | 1GPU | 2GPU | 4GPU  |
> | ------------------------- | ---- | ---- | ----- |
> | SplitGPU                  | N/A  |0.08  | 0.119 |
> | SameGPU (homo batching)   | 0.067| 0.126| 0.252 |
> | Disaggregated vLLM        | N/A  |0.118 | 0.276 |
> | **Orthrus (ours)**        | **0.075**|**0.142** | **0.288** |
>
> - Generation throughput (requests/sec): the higher the better
>
> |                         | 1GPU | 2GPU | 4GPU  |
> | ----------------------- | ---- | ---- | ----- |
> | SplitGPU                | N/A  |0.11  | 0.141 |
> | SameGPU (homo batching) |0.08  |0.148 | 0.296 |
> | Disaggregated vLLM      | N/A  |0.14  | 0.327 |
> | **Orthrus (ours)**      |**0.088** |**0.166** | **0.336** |
>
>
> **Evaluation Setup (incorporating W2)**
> This new experiment uses variable-length prompts (avg 500 tokens, max 2000 tokens), variable and long decode lengths (avg 3000 tokens, max 4000 tokens), multi-document retrieval (1–10 documents), and re-embedding phases. Orthrus maintains the highest throughput under these conditions (928 decode tokens/s and 105 embed tokens/s, or equivalently, 0.288 embedding request/sec and 0.336 generation request/sec) compared to all baselines.
>
>
> ### (W2, W5) Synthetic and narrow workloads
>
> **New Experiments**
> To address the concern, we now include Iter-RetGen retrieval (Shao et al., 2023) running on the 2WikiMultihopQA benchmark dataset. Specifically, embedding and generation alternates, which are then followed by a final long generation step. The evaluation results are presented in W1. We will, of course, add this new result to our manuscript.

---

> ### Author Response · Authors · 2025-11-20
> **Response to Reviewer PhM6 (2/2)**
>
> ### (W3, Q3) Sensitivity analysis regarding “fill residual with embeddings” assumption
>
> We acknowledge that the analytical model assumes proportional scaling and that embeddings generally fit in residual slots. The implementation does not depend on this assumption. Orthrus opportunistically schedules embeddings when memory is available, and falls back to pure generation iterations when decode lengths are long or KV cache fragmentation prevents embedding placement. We will clarify this in the final version.
>
>
> ### (W4) Missing system details
>
> Orthrus does not fuse or replace any core operators. FlashAttention, sampling, LoRA application, and paging are inherited directly from vLLM, which is our intention for streamlined code merge in the future. The incremental pooling operator is lightweight (<1 ms) and does not affect the forward or decode path. Punica/S-LoRA-style fused kernels are complementary; Orthrus can benefit from them but does not rely on them. We will make this clearer.

---

### Author Response · Authors · 2025-12-03
**Author Summary**

We present a specialized inference kernel for modern RAG. This technique merits wider visibility on the basis of its intellectual contributions and broader impacts:

- **Technical Novelty**: This is the first work that schedules both embedding and generation within each batch, thus called *heterogeneous batching* (Section 3 of manuscript). Our technique effectively marries compute-bound embedding and memory-bound generation within each iteration (Section 3.1 of manuscript), delivering higher total throughput and lower latency compared to state-of-the-art techniques used in today's LLM serving stack, including disaggregated execution ([Response to Reviewer PhM6(1/2)](https://openreview.net/forum?id=aWF6wVsGYS&noteId=BgqppjAlRl)). That is, ***our approach is well-motivated, unorthodox, and comprehensively tested.***
- **High-quality Implementation/Evaluation**: Unlike previous works, ***we integrate our method inside production-grade inference engine (i.e., vLLM),*** (Section 3.1 of manuscript). Moreover, we have re-implemented all baselines inside the same framework for fairness. Our evaluation shows higher performance for a wide range of models and RAG workloads  (Section 4 of manuscript, [Response to Reviewer PhM6](https://openreview.net/forum?id=aWF6wVsGYS&noteId=BgqppjAlRl)). Ours offers higher performance advantage when resource is more constrained ([Response to Reviewer PhM6](https://openreview.net/forum?id=aWF6wVsGYS&noteId=BgqppjAlRl)).
- **Timely**: Modern AI is extremely resource-hungry. Our work tackles it by taking a systems approach: no approximation, just sophistigated resource usage. Such research is often most successful when a simple intuition is validated through deep engineering, which we prioritize in this work and is recognized by reviwers.


## Addressing Reviewers' Concerns

Within the one-week rebuttal period, we could respond to all reviewers' concerns with concrete numbers, addressing their concerns. This was only possible because we already had a well-designed kernel that can easily generalize to various workloads and computational environments. More concretely:

- **Additional Baselines**: We have newly compared to disaggregated inference (i.e., TetriInfer, ShuffleInfer), using 1, 2, and 4 GPUs. Our approach was faster. ([Response to Reviewer PhM6 (1/2)](https://openreview.net/forum?id=aWF6wVsGYS&noteId=BgqppjAlRl))
- **Additional Workloads**: Besides controlled experiments, we have additionally included a RAG pipeline (i.e., Iter-RetGen on 2WikiMultihopQA), featuring varying iteration counts and generation lengths. Our performance advantages still held. ([Response to Reviewer PhM6 (1/2)](https://openreview.net/forum?id=aWF6wVsGYS&noteId=BgqppjAlRl))
- **What if no params are shared between embedding and generation**: Our new experiments still demonstrated higher performance even with two completely separate models (Qwen-0.6B models). ([Response to Reviewer 7SYe (1/2)](https://openreview.net/forum?id=aWF6wVsGYS&noteId=dVCgZfaKf0))

In summary, we investigated an unconventional technique—heterogeneous batching—specifically designed for modern RAG. Its performance gains are demonstrated in comparison to state-of-the-art approaches. This work expands our knowledge.

---

### Meta-Review · Area_Chair_WawC · 2025-12-21

**Summary:**

Reviewers agree that the problem is timely, the system design is clean and compatible with production-grade infrastructure, and the empirical gains, under the evaluated workload, are substantial and practically meaningful.

At the same time, the reviews identify substantial limitations. Several closely related systems (TetriInfer, ShuffleInfer, MACE, SageServe, recent vLLM deployments) already support heterogeneous or mixed-workload batching, but the paper does not compare against them, making the novelty and the strength of the claimed 1.28-4.52× gains less clear. The evaluation uses synthetic traces, fixed-length inputs, single-model workloads, and simplified embedding/generation ratios that do not resemble realistic RAG patterns, leaving uncertainty about generality. Some theoretical assumptions (proportional compute/memory costs, consistent residual memory availability) appear optimistic, and empirical sensitivity analyses are missing. Certain baselines are implemented in ways that are easily improved, and some results appear contradictory across figures. Several reviewers also question whether the framework holds when embeddings and generators are different models or sizes, which is common in practice.

**Reviewer Concerns:**

Authors have included additional comparisons with baselines and workloads to validate its generalizability. However, concerns of theoretical analyses and validity of assumptions are still outstanding. In general, the paper needs major revisions in experiments and writings.

**Reviewer Scores:**

4,4,4,6

---

### Decision · Program_Chairs · 2026-01-26

Reject